Journal of Machine Learning Research (2025) 1-11         Submitted ; Published

# Effortless Vision-Language Model Specialization in Histopathology without Annotation

**Jingna Qiu**                         JINGNA.QIU@FAU.DE

**Nishanth Jain**                   NISHANTH.JAIN@FAU.DE
*Friedrich-Alexander-Universität Erlangen-Nürnberg, Erlangen, Germany*

**Jonas Ammeling**               JONAS.AMMELING@THI.DE
*Ingolstadt University of Applied Sciences. Ingolstadt, Germany*

**Marc Aubreville**         MARC.AUBREVILLE@HS-FLENSBURG.DE
*Flensburg University of Applied Sciences, Flensburg, Germany*

**Katharina Breininger**      KATHARINA.BREININGER@UNI-WUERZBURG.DE
*Julius-Maximilians-Universität Würzburg, Würzburg, Germany*

**Editor:**

## Abstract

Recent advances in Vision-Language Models (VLMs) in histopathology, such as CONCH and QuiltNet, have demonstrated impressive zero-shot classification capabilities across various tasks. However, their general-purpose design may lead to suboptimal performance in specific downstream applications. While supervised fine-tuning methods address this issue, they require manually labeled samples for adaptation. This paper investigates annotation-free adaptation of VLMs through continued pretraining on domain- and task-relevant image-caption pairs extracted from existing databases. Our experiments on two VLMs, CONCH and QuiltNet, across three downstream tasks reveal that these pairs substantially enhance both zero-shot and few-shot performance. Notably, with larger training sizes, continued pretraining matches the performance of few-shot methods while eliminating manual labeling. Its effectiveness, task-agnostic design, and annotation-free workflow make it a promising pathway for adapting VLMs to new histopathology tasks. Code is available at https://github.com/DeepMicroscopy/Annotation-free-VLM-specialization.

**Keywords:** vision-language models, histopathology, task adaptation

## 1 Introduction

Vision-Language Models (VLMs) integrate images and textual descriptions to improve representation learning. Their success on natural image datasets has catalyzed their adaptation for histopathology image analysis. Several specialized VLMs tailored to histopathology have been developed, including PLIP (Huang et al. (2023)), QuiltNet (Ikezogwo et al. (2023)), and CONCH (Lu et al. (2024)). PLIP and QuiltNet fine-tune CLIP (Radford et al. (2021)), with PLIP utilizing image-caption pairs sourced from Twitter discussions among pathologists, while QuiltNet leverages educational videos from YouTube and medical literature. On the other hand, CONCH builds on CoCa (Yu et al. (2022)), incorporating a captioning objective alongside CLIP's contrastive objectives and being trained on a dataset derived from PubMed articles and educational materials. Thanks to extensive and varied pretraining,

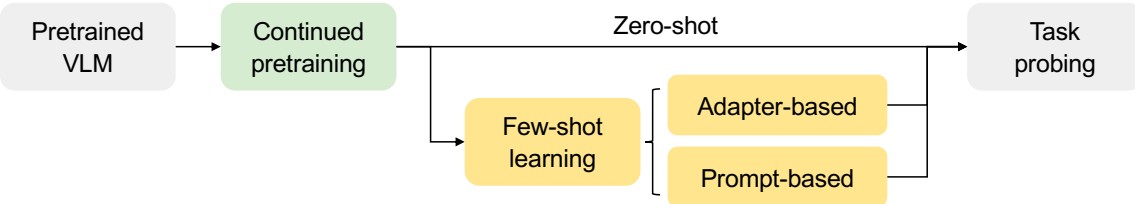

Figure 1: Retrieved image-caption pairs are utilized in the continued pretraining of a pretrained VLM using a contrastive loss. Their effectiveness is evaluated in both zero-shot and few-shot learning scenarios, in comparison to the original VLM.

these models exhibit robust zero-shot performance in histopathology image interpretation, including tile classification and downstream tasks like Whole Slide Image (WSI) segmentation, achieved by aggregating tile-level results.

Nevertheless, training on broad and heterogeneous datasets may hinder a model's effectiveness on specialized tasks that necessitate more focused representations. To mitigate this issue, various supervised fine-tuning approaches have been proposed to refine model adaptation. Full fine-tuning updates all parameters of a pretrained encoder along with a classifier on top, while partial fine-tuning targets selective layers. Parameter Efficient Fine-Tuning (PEFT) methods introduce lightweight modules (e.g., adapters) that train only these components, keeping the pretrained weights fixed, thereby enhancing computational efficiency. Among these, LoRA (Hu et al. (2022)) integrates trainable low-rank matrices in parallel with frozen dense layers to capture the residual for adjusting the original layer outputs. Adapter-based PEFT strategies, such as CLIP-Adapter (Gao et al. (2024)), employ adapters that utilize the pre-trained image and text features and blend the adapter output as a residual to form the final features. Prompt-based adaptations, like Context Optimization (CoOp) (Zhou et al. (2022b)), involve training learnable prompt vectors instead of relying on hand-crafted prompts.

However, all these methods necessitate manually annotated data for supervision, with PEFT methods being relatively data-efficient and often employing few-shot datasets. Curating a labeled dataset requires specialized knowledge in pathology and becomes particularly challenging when identifying samples for rare diseases. This motivates our exploration of utilizing image-caption pairs from existing histopathology databases to enhance foundation model adaptation without the need for annotations. Specifically, we identify domain- and task-relevant samples through string matching. In the context of breast cancer classification, domain-relevant image-caption pairs are those where the caption includes the organ term "breast", while task-relevant pairs are a subset of the domain-specific pairs that contain one of the class names (e.g., "normal", "benign", "in situ", and "invasive"). These domain- and task-specific pairs are subsequently used to continue the pretraining of the VLM using a contrastive loss, referred to as Domain-adaptive PreTraining (DAPT) and Task-adaptive PreTraining (TAPT), respectively.

Our evaluation involves conducting DAPT and TAPT through full parameter updates and assessing the adapted model's performance on the targeted task in comparison to the original model. Furthermore, we investigate whether continued pretraining benefits

subsequent few-shot learning methods, specifically CoOp. Our experiments, encompassing three pathology tasks, including breast cancer classification on BACH (Aresta et al. (2019)), colorectal polyp classification on MHIST (Wei et al. (2021)), and prostate cancer grading on SICAP (Silva-Rodríguez et al. (2020)), demonstrate that the image-caption pairs identified from the retrieval dataset Quilt1M (Ikezogwo et al. (2023)) significantly enhance both zero-shot and few-shot performance of the foundation model in downstream tasks.

## 2 Related Work

**Continued Pretraining:** The methods of DAPT and TAPT have been evaluated using the language model RoBERTa (Liu et al. (2019)) in previous work (Gururangan et al. (2020)), where relevant datasets were utilized as domain-specific data (e.g., Amazon reviews for the target task of IMDB review sentiment classification), while unlabeled training data (curated alongside the target task training set) served as task-specific data. In contrast, our work investigates the impact of retrieved image-caption pairs on VLM adaptation, employing organ and class names as domain- and task-specific data, respectively.

**VLM Adaptation via Fine-Tuning:** Prior research has predominantly focused on PEFT, particularly using few-shot datasets. Among **adapter-based** methods, CLIP-Adapter (Gao et al. (2024)) introduces an adapter consisting of two linear layers atop the text and image encoders, respectively, blending the adapter's output with the original embeddings to generate the final output. Tip-adapter (Zhang et al. (2022)) is a cache-based adapter that calculates image embeddings for each few-shot training sample, integrating their one-hot labels weighted by their similarity to the input image during classification. Tip-X (Udandarao et al. (2023)) further incorporates image-text similarities, given that the contrastive loss aims to align the two modalities. The cache modules in both Tip-adapter and Tip-X are fine-tuned using the few-shot training data. Among **prompt-based** methods, CoOp (Zhou et al. (2022b)) transforms context words in a prompt into a set of learnable vectors, replacing hand-crafted prompts such as "A photo of {class}". Meanwhile, CoCoOp (Zhou et al. (2022a)) extends CoOp by implementing a lightweight module that generates an input-conditional token for each image. Additionally, CLIP-LoRA (Zanella and Ben Ayed (2024)) applies Low-Rank Adaptation (LoRA) to CLIP by injecting low-rank matrices into the query, key, and value matrices within the attention block, maintaining a rank of 2.

## 3 Method

Our goal is to evaluate the effectiveness of retrieved image-caption pairs in adapting pretrained histopathology VLMs to specific downstream tasks. These pairs are utilized to continue pretraining a foundational VLM using a contrastive loss. We compare the performance of the adapted model to that of the original model in both zero-shot and few-shot learning scenarios. The workflow is illustrated in Fig. 1.

### 3.1 Image-Caption Pair Retrieval

To conduct DAPT and TAPT, we collect domain- and task-specific image-caption pairs via string matching from existing histopathology image-caption databases. The aim is to determine whether adapting the model by exposing it to general domain information (e.g.,

regarding organs) or more specialized information on targeted classes is more beneficial. Domain-specific image-caption pairs are identified if the caption includes one of the site keywords, such as the name of the organ. In specific cases where a certain subject is involved in the task, such as in colorectal polyp classification, the subject "polyp" also serves as a keyword. Task-specific pairs are filtered as a subset of the domain-specific pairs using class names pertinent to the task. Synonyms and alternative terms, such as Ductal Carcinoma In Situ (DCIS) and Lobular Carcinoma In Situ (LCIS) for the class "in situ" in breast cancer classification, are excluded to avoid reliance on specialized medical knowledge. Full names are employed when class names involve abbreviations.

Once identified, the pairs are ranked based on the alignment of image-caption pairs using the CONCH model due to its general high performance in zero-shot classification among histopathology tasks. Alignment scores are computed as the cosine similarity between normalized image embeddings $x_i$ and caption embeddings $y_i$ for the $i^{\text{th}}$ pair, as

$$sim(x_i, y_i) = x_i^\top y_i. \tag{1}$$

This ranking prioritizes high-quality pairs when using a limited training set and facilitates filtering out noisy data by discarding poorly related pairs.

We opted for a straightforward string matching method, as preliminary experiments showed that it leads to higher-quality retrievals than similarity searches based on embeddings from PathologyBERT (Santos et al. (2023)) or caption classification using Gemma3 (Kamath et al. (2025)).

## 3.2 Continued Pretraining

We proceed with continued pretraining using the domain-specific image-caption pairs in DAPT and the task-specific pairs in TAPT. Both the image and text encoders in the VLM are updated using a dual-encoder contrastive loss (Yu et al. (2022)), as

$$L_{\text{contrast}} = -\frac{1}{N} \left( \sum_{i=1}^{N} \log \frac{\exp(x_i^\top y_i)}{\sum_{j=1}^{N} \exp(x_i^\top y_j)} + \sum_{i=1}^{N} \log \frac{\exp(y_i^\top x_i)}{\sum_{j=1}^{N} \exp(y_i^\top x_j)} \right), \tag{2}$$

where $N$ is the batch size. The cosine similarities of the paired embeddings are maximized relative to other negative pairings within the batch.

## 3.3 Evaluation of Continued Pretraining Effect

To assess the impact of continued pretraining, we evaluate the model in both zero-shot and few-shot scenarios. In zero-shot classification, the text encoder computes embeddings for each class text prompt, and the class with the highest cosine similarity is assigned to the test image. We utilize the same multiple class descriptions and prompt templates employed in CONCH (Lu et al. (2024)) for prompt ensembling, ensuring fair comparisons.

For few-shot learning, we adopt CoOp due to its established effectiveness. In CoOp, prompt vectors are learned to substitute hand-crafted prompts for the text encoder input, while the pretrained text encoder remains frozen. We conduct experiments with two types of prompts: a unified context prompt $[V]_1[V]_2[V]_3 \ldots [V]_M[CLASS]$ and a Class-Specific Context (CSC) prompt $[V]_1^c[V]_2^c[V]_3^c \ldots [V]_M^c$ unique to each class $c$. Here, $[V]_M$ denotes

Table 1: Keywords for domain- and task-specific image-caption pairs retrieval and the retrieved pair amounts for DAPT and TAPT.

| Task | Site keywords | Class keywords | DAPT | TAPT |
|------|---------------|----------------|------|------|
| BACH | breast | normal, benign, in situ, invasive | 5437 | 896 |
| MHIST | colon, colorectal, polyp | hyperplastic, benign, sessile, serrated, adenoma | 5224 | 806 |
| SICAP | prostate, gland | non-cancerous, Gleason | 10749 | 154 |

a context token of the same dimension as the word embeddings and $M$ is the context length. It has been shown that unified and CSC prompts yield better performance in generic and fine-grained object classification on natural images, respectively, and shorter context lengths enhance generalization while longer ones improve performance (Zhou et al. (2022b)). Following the original paper, we experiment with both context types and lengths of 4 and 16 to determine the optimal combination for histopathology tasks.

## 4 Experimental Setups

### 4.1 Image-caption Source Database Quilt1M

The Quilt1M database (Ikezogwo et al. (2023)) contains $1,017,708$ image-caption pairs collected from multiple sources: $802,144$ pairs from $1,087$ hours of educational histopathology YouTube videos, $59,371$ pairs from PubMed open-access articles, $22,682$ histopathology-related pairs from the LAION-5B dataset (Schuhmann et al. (2022)), and $133,511$ pairs from $55,000$ curated tweets in OpenPath (Huang et al. (2023)). Extracting image-caption pairs from YouTube videos presents challenges such as accurate speech-to-text conversion, particularly for medical terminology, frame extraction and precise alignment of image frames with corresponding text. Due to the dataset's scale, manual verification is impractical. Aubreville et al. (Aubreville et al. (2024)) refined Quilt1M by removing low-quality images and those containing extraneous elements such as narrators or overlaid text. This process resulted in a cleaned subset of $232,039$ image-caption pairs, which we use in this study.

### 4.2 Histopathology-specific Vision-language Models

**CONCH** is a CoCa-based VLM trained on an in-house dataset of over 1.17 million image-caption pairs sourced from PubMed and educational materials. **QuiltNet** is a CLIP-based VLM trained on the Quilt1M database. The selection of these two models aims to understand the impact of whether the retrieval dataset Quilt1M was included in the original foundation model's pretraining process.

### 4.3 Downstream Tasks and Datasets

**Breast Cancer Classification on BACH** The BACH dataset (Aresta et al. (2019)) contains microscopy images used for a breast cancer subtyping task across four classes: normal, benign, in situ carcinoma, and invasive carcinoma. The dataset comprises 400

images, with 100 images per class. Each image has a standardized resolution of $2048 \times 1536$ pixels at $0.42 \frac{\mu m}{px}$.

**Colorectal Polyps Classification on MHIST** The MHIST dataset (Wei et al. (2021)) comprises 3152 images (2162 Hyperplastic Polyps (HPs), 990 Sessile Serrated Adenomas (SSAs)) extracted from 328 WSIs, each with a resolution of $224 \times 224$ pixels at $8\times$ magnification. The test set of 977 images are used for evaluation.

**Prostate Cancer Grading on SICAP** The SICAP dataset (Silva-Rodríguez et al. (2020)) is used for prostate cancer diagnosis based on the Gleason grading system (Gleason (1992)). SICAP contains 10340 image patches extracted from 182 WSIs (4417 non-cancerous, 1636 grade 3, 3622 grade 4, 665 grade 5). Each patch has a resolution of $512 \times 512$ pixels at $10\times$ magnification. The test set of 2122 images are used for evaluation.

A complete list of keywords used to identify relevant domain- and task-specific image-caption pairs for all datasets is provided in Table 1.

### 4.4 Evaluation Metrics and Implementation Details

Balanced accuracy is used to evaluate the experiments on BACH and MHIST, while Cohen's quadratic kappa is used for SICAP, following (Lu et al. (2024)).

To assess the impact of training size $N$ on continued pretraining, we select the first $N$ pairs in the sorted retrievals (according to their CONCH alignment score). We adopt the convention of using "shots" to denote the training size, defined as $N = shots \times num\_classes$, following few-shot learning conventions. Continued pretraining experiments perform full parameter update and use contrastive loss. Few-shot experiments perform CoOp and use cross-entropy loss. All experiments use the AdamW optimizer with cosine annealing as the learning rate scheduler for 50 epochs, with initial learning rate and weight decay parameters being tuned for each training size based on the minimal training loss after 5 epochs.

## 5 Results and Discussion

### 5.1 Effect of Continued Pretraining on Zero-shot

Figure 2 compares the performance of DAPT and TAPT across various training sizes. DAPT (blue) demonstrates consistent improvements in zero-shot performance across all tasks and models compared to the original unadapted model (black dash line), except for MHIST when applied to QuiltNet. The benefit of increasing the number of training image-caption pairs on DAPT is not pronounced, except in the case of MHIST when applied to CONCH. Moreover, performance may decline after a certain amount of training data is used, as observed on SICAP. Given that the training pairs are sorted by alignment score according to CONCH, it is likely that lower-ranked pairs may detrimentally affect model performance. In contrast, TAPT (red) achieves more stable improvements as additional training pairs are utilized. When applied to QuiltNet, TAPT surpasses DAPT, enhancing the original model's performance by 79.62% and 15.22% on BACH and MHIST, respectively, compared to the 55.34% and 10.41% improvements achieved by DAPT. Furthermore, TAPT significantly increases the weighted Cohen's kappa on SICAP from 0.02 to 0.50, achieving higher performance improvement than DAPT with fewer training pairs. Interestingly, when moving to CONCH, DAPT exhibits higher and more consistent performance improvements across

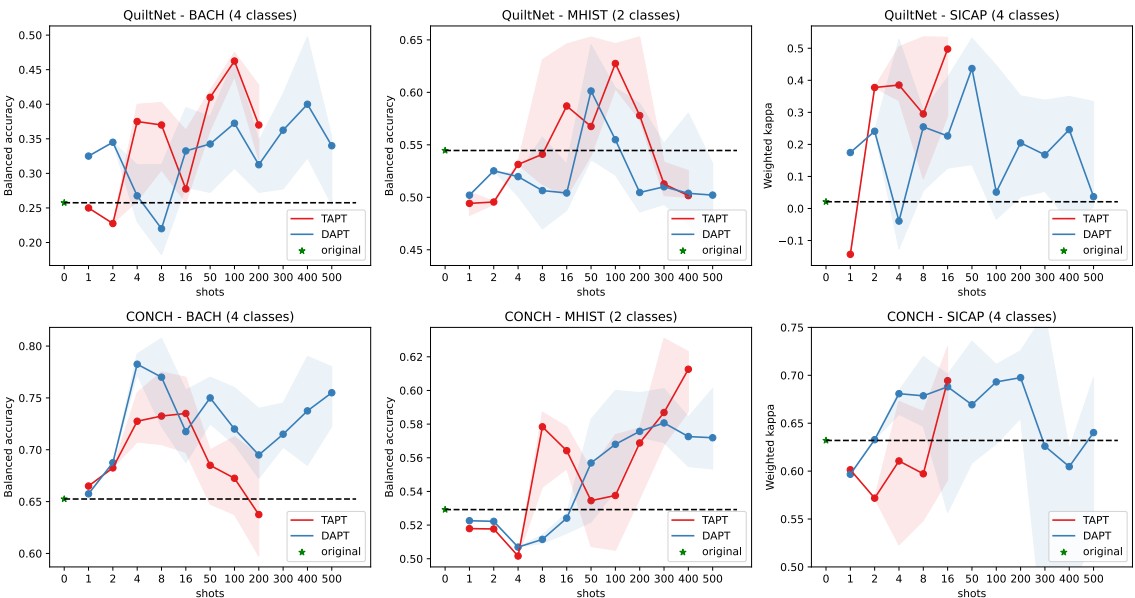

Figure 2: Comparison of continued pretraining methods, DAPT and TAPT, with the original unadapted model on QuiltNet (top row) and CONCH (bottom row) across three pathology tasks up to 500 shots. The performance of TAPT may terminate earlier if insufficient pairs are retrieved. The results show the median values from five repetitions, with minimum and maximum values shaded.

datasets, especially on BACH and SICAP, suggesting that domain adaptation via large-scale caption-text alignment benefits CONCH more strongly. This difference may arise because QuiltNet has already been pre-trained on Quilt1M (Ikezogwo et al. (2023)), making domain information in the training pairs redundant, whereas CONCH, without prior exposure to Quilt1M, benefits more from DAPT. Overall, the use of retrieved domain- or task-specific image-text pairs demonstrates promising potential for model adaptation, yielding more substantial improvements particularly when the baseline performance of the original VLM is low. This is especially noteworthy given that the retrieval database, Quilt1M, is primarily sourced from uncurated YouTube videos and lacks manual verification, which underscores the robustness of the adaptation process of continued pretraining even in the presence of noisy retrieval sources. Additional results utilizing LoRA for continued pretraining as a replacement for full parameter updates, along with the use of PathologyBERT and Gemma3 models for assigning pseudo-labels to the retrieved pairs to balance the training set, are presented in the supplementary material.

## 5.2 Effect of Continued Pretraining on Few-shot

Due to the relatively poorer performance improvements of DAPT and TAPT on MHIST with QuiltNet in zero-shot evaluations compared to other datasets and models, we apply the few-shot method CoOp to assess whether the adapted model's performance aligns with

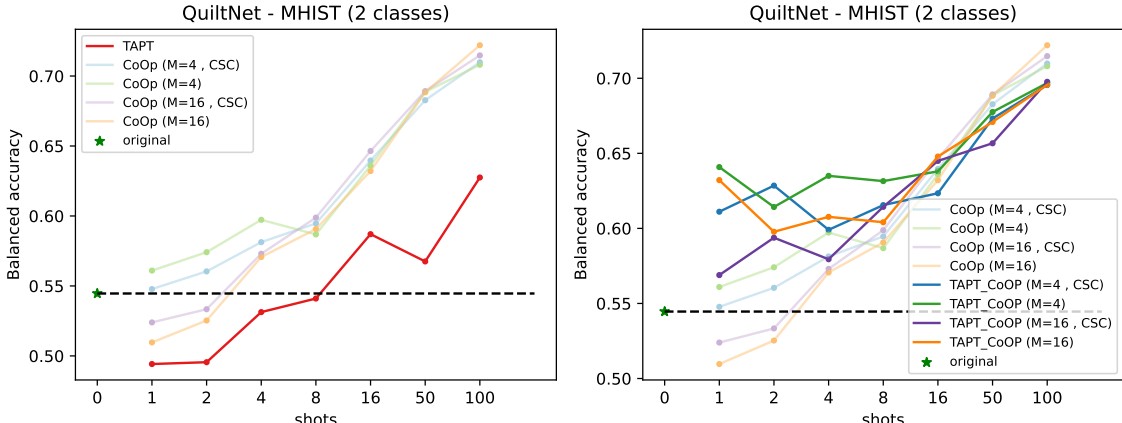

Figure 3: Left: Comparison of TAPT and CoOp using unlabeled and labeled training data, respectively. $M$ denotes the prompt's context length. CSC indicates the use of class-specific context, while the unified context is used otherwise. Right: Comparison of CoOp with and without TAPT across various context types and lengths. Results present the median values from ten randomly selected few-shot sets.

that of few-shot learning. Additionally, we investigate whether applying CoOp on top of the adapted model yields higher performance than the original unadapted model. We repeat with 10 randomly selected few-shot sets from the MHIST training set and report the median values from the evaluation on the test set (see Fig. 3). Our observations indicate that a context length of 4 with a unified context prompt performs the best up to 4 shots for CoOp on MHIST, showing minimal differences thereafter with varying context lengths and types. Notably, the performance improvements to the original VLM achieved through CoOp can also be attained with TAPT when a larger training size with no need for manual annotation. For instance, TAPT achieves performance comparable to CoOp at 8 shots by utilizing 16 shots, and a performance similar to CoOp at 16 shots with 100 shots (Fig. 3 left). Further, with up to 8 shots, TAPT significantly enhances CoOp by achieving higher performance compared to its application on the original unadapted model (Fig. 3 right).

## 6 Conclusion and Future Work

We demonstrated that continued pretraining of a VLM with retrieved domain- and task-specific image–caption pairs enhances its performance in both zero-shot and few-shot learning settings. Our method achieves results comparable to the few-shot learning approach CoOp when trained on larger datasets, while eliminating the need for manual annotation. Its label-free and task-agnostic design offers a promising direction for adapting foundation VLMs to new tasks.

Future work includes optimizing batch composition strategies, such as avoiding batches containing images with highly similar texture descriptions, to stabilize contrastive loss convergence. Applying a threshold on CONCH alignment scores could filter out poorly aligned

pairs and prevent performance degradation. While CONCH scores effectively proxy pair quality, ensembling alignment measurements from multiple VLMs may further enhance data selection. Integrating our annotation-free adaptation with few-shot learning by ranking pairs based on similarity to few-shot images could improve training relevance and facilitate more meaningful clinical deployment.

## Acknowledgments and Disclosure of Funding

We acknowledge support by d.hip campus - Bavarian aim (J.Q.), the German Research Foundation (DFG) project 460333672 CRC1540 EBM, project 405969122 FOR2886 Pandora, as well as the scientific support and HPC resources provided by the Erlangen National High Performance Computing Center (NHR@FAU) of the Friedrich-Alexander-Universität Erlangen-Nürnberg (FAU). NHR funding is provided by federal and Bavarian state authorities. NHR@FAU hardware is partially funded by the German Research Foundation (DFG) – 440719683.

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

# Effortless Vision-Language Model Specialization in Histopathology without Annotation

**Jingna Qiu**                  JINGNA.QIU@FAU.DE

**Nishanth Jain**                 NISHANTH.JAIN@FAU.DE
*Friedrich-Alexander-Universität Erlangen-Nürnberg, Erlangen, Germany*

**Jonas Ammeling**                JONAS.AMMELING@THI.DE
*Ingolstadt University of Applied Sciences. Ingolstadt, Germany*

**Marc Aubreville**          MARC.AUBREVILLE@HS-FLENSBURG.DE
*Flensburg University of Applied Sciences, Flensburg, Germany*

**Katharina Breininger**    KATHARINA.BREININGER@UNI-WUERZBURG.DE
*Julius-Maximilians-Universität Würzburg, Würzburg, Germany*

**Editor:** My editor

## 1 Supplementary Materials

We investigate the effect of balancing the training set among classes on QuiltNet by leveraging PathologyBERT and Gemma3 models for assigning pseudo-labels. As shown in Fig. S1 (top row), we do not observe significant improvements from balancing, nor do we see better performance with Gemma3 classifications, which have a higher correctness rate compared to the labels provided by PathologyBERT. Additionally, we utilize Low-Rank Adaptation (LoRA) for continued pretraining as a replacement for full parameter updates to evaluate the generalization of the retrieved image-caption pairs. As shown in Fig. S1 (bottom row), we found similar patterns of performance improvement compared to full parameter update. However, a lower rank (e.g., 2) can lead to decreased performance compared to full parameter update when more training data is employed.

We provide supplementary figure for Fig. 2 with normalized scale to easier performance comparisons between QuiltNet and CONCH in Fig. S2.

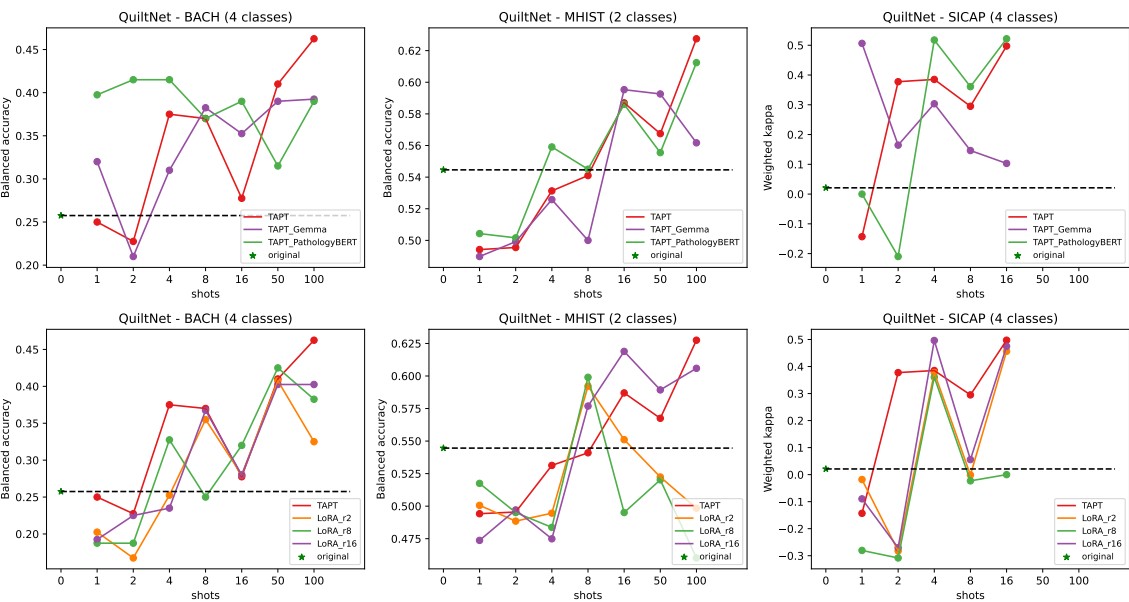

Figure S1: Top row: Effect of leveraging PathologyBERT and Gemma3 for balancing the training set. Bottom row: Effect of replacing full parameter update with LoRA.

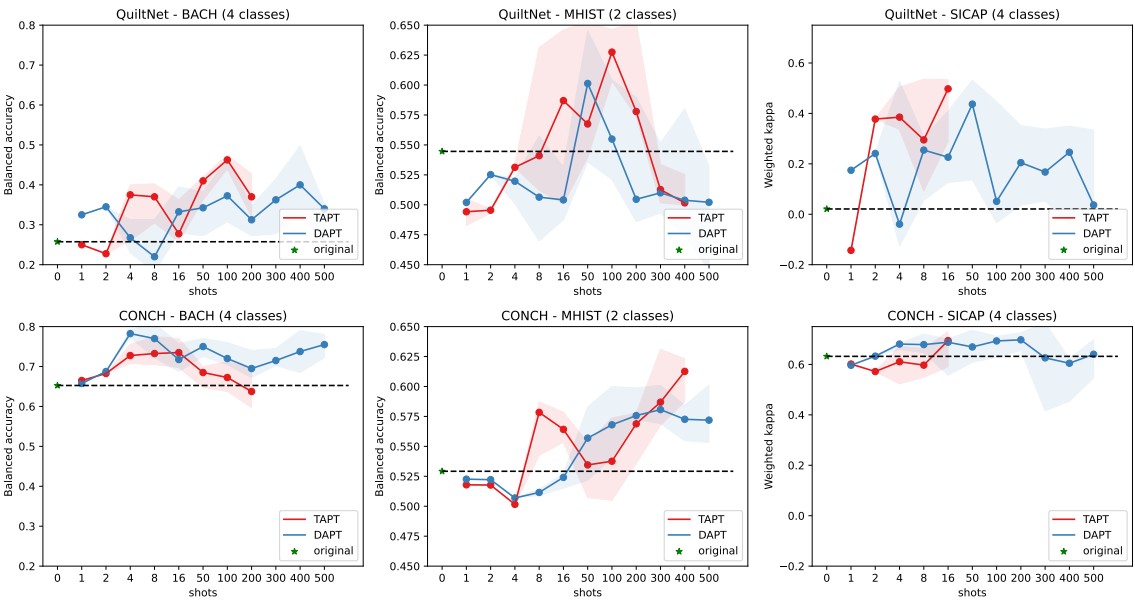

Figure S2: Supplementary figure for Fig. 2 with normalized scale.

