# OpenReview forum: "Effortless Vision-Language Model Specialization in Histopathology without Annotation"
_MICCAI.org/2025/Workshop/COMPAYL — COMPAYL 2025_

### Official Review · Reviewer_8kjR · 2025-07-11
**Annotation-Free VLM Specialization for Histopathology: A Promising Approach with Room for Deeper Analysis**

**Rating:** 4
**Confidence:** 4

**Review:**

Short summary: This paper explores an annotation-free method for adapting Vision-Language Models (VLMs) to specific histopathology tasks through continued pretraining on domain- and task-relevant image-caption pairs. The authors introduce Domain-adaptive PreTraining (DAPT) and Task-adaptive PreTraining (TAPT) and demonstrate their effectiveness in enhancing both zero-shot and few-shot performance on histopathology datasets like BACH, MHIST, and SICAP. A notable finding is that this annotation-free approach can achieve performance comparable to supervised few-shot methods like CoOp with a larger training size.

Strengths: Novelty of Annotation-Free Adaptation: The paper addresses a significant challenge in histopathology, which is the scarcity and cost of manual annotations, by proposing an annotation-free adaptation method for VLMs. This is a key strength as it offers a practical solution for leveraging existing data; Comprehensive Evaluation: The study evaluates two different VLMs (QuiltNet and CONCH) across three distinct histopathology tasks (breast cancer classification, colorectal polyp classification, and prostate cancer grading), providing a robust assessment of the proposed methods; Demonstrated Effectiveness: The experiments clearly show that continued pretraining with retrieved image-caption pairs significantly enhances both zero-shot and few-shot performance of VLMs in histopathology, particularly when the baseline VLM performance is low; Comparability to Few-Shot Learning: The finding that the annotation-free TAPT can achieve performance comparable to the data-efficient CoOp method with larger training sizes is a strong result, highlighting the potential of this approach to reduce annotation burden.

Weaknesses: Limited Analysis of Performance Decline: While the paper mentions that performance may decline after a certain amount of training data, particularly for DAPT on SICAP, and attributes it to lower-ranked (noisy) pairs, a more in-depth analysis of this phenomenon would be beneficial. Understanding the optimal training size and strategies to mitigate the negative impact of lower-quality data is crucial; Clarity on "Shots" Definition in DAPT/TAPT: The paper uses "shots" to denote training size (N = shots * num_classes) following few-shot learning conventions. While this makes sense for comparison with few-shot methods, it could be clearer what constitutes a "shot" in the context of DAPT/TAPT, where there are no explicit "classes" being sampled for labeled examples. More specifically, how N is determined for DAPT/TAPT to align with the few-shot "shots" concept could be elaborated; Depth of Discussion on Model Differences: The paper touches upon why CONCH benefits more from DAPT than QuiltNet (due to QuiltNet already being pretrained on Quilt1M). However, a more detailed discussion on the architectural differences between CONCH (CoCa-based) and QuiltNet (CLIP-based) and how these might influence the effectiveness of DAPT/TAPT would strengthen the analysis.

Detailed comments: Evaluation Metrics (Section 4.4): The choice of balanced accuracy for BACH and MHIST, and Cohen's quadratic kappa for SICAP, is justified by citing previous work. This maintains consistency and allows for fair comparisons; Figure 3 (Few-shot analysis): The left graph clearly shows TAPT's ability to match CoOp's performance with more data, which is the core argument for annotation-free adaptation. The right graph further reinforces the benefit of TAPT when combined with CoOp, demonstrating a synergistic effect.

---

### Official Review · Reviewer_yunR · 2025-07-14
**Valuable Empirical Insights on VLM Adaptation and Data Quality Effects with Opportunities for Enhanced Critical Discussion**

**Rating:** 4
**Confidence:** 4

**Review:**

Contribution:
The authors propose an annotation-free method to adapt pre-trained vision-language models for histopathology to specific downstream subdomains and tasks using continued pretraining on carefully selected image-caption pairs. Based on the Quilt1M dataset, domain or task-specific image-caption pairs are selected using string matching. Afterwards these pairs are ranked according to the embedding similarity between the image and caption within the pair, serving as a proxy for data quality. These pairs are then used to fine-tune CONCH and QuiltNet, showing that zero-shot classification generally improved compared to the untuned model.

Strengths:
- The study uses established publicly available models and datasets, which makes it highly reproducible. Also, multiple datasets are used which helps us understand more general trends across domains.
- The study also shows empirically that zero-shot classification generally improves using the selected data, which for a large part come from uncurated YouTube videos. This is valuable information for other researchers looking to train a VLM using the same data.
- The study shows that just using more data for fine-tuning does not consistently improves the zero-shot performance where sometimes the performance dips under the untuned model’s. Instead, the quality or representativeness of the selected samples seems to be more important which can inform future work.
- The authors indeed generally show that zero-shot classification can improve without additionally labeled data like is needed for CoOp, even if supervised methods are still superior. Therefore the study shows incremental improvements toward tackling the reliance on annotated data within the field of computational pathology.

Weaknesses:
- The paper would benefit from a more thorough discussion of the method's limitations, particularly given the concerning finding that model performance sometimes degrades when using more training samples, which is a counterintuitive result that suggests fundamental issues with the data selection or ranking approach. Although the ranking method for the selected image-caption pairs sounds intuitive, using it as a proxy for data quality for training samples might not be accurate. You would want to select image-caption pairs that represent your underlying data distribution well so that the fine-tuned model generalizes well, but in this case the cosine similarity between CONCH embeddings only measures how well aligned the image and caption are according to CONCH's existing representation space, not how representative or useful those pairs are for learning the target task. This could lead to selecting highly correlated but potentially redundant or biased samples that don't capture the full complexity of the target domain, which may explain why performance sometimes degrades with more training data as lower-ranked pairs are added. Despite this, it provides potential for future researchers to find better options for data selection and/or ranking. This acknowledgement, together with a concrete proposal for improved methods, could have benefitted the paper greatly.

---

### Official Review · Reviewer_ybmb · 2025-07-14
**A Vision-Language Model (VLM) specialization approach for histopathology that builds existing VLM models to perform annotation-free adaptation for downstream tasks**

**Rating:** 4
**Confidence:** 5

**Review:**

In this study, the authors propose a Vision-Language Model (VLM) specialization approach for histopathology that builds existing VLM models to perform annotation-free adaptation for downstream tasks. The authors use the publicly available CONCH model to achieve domain-adaptive pretraining (DAPT – organ-specific) and task-adaptive pretraining (TAPT – in situ, invasive) and extract image-text pairs using string matching and similarity thresholding to select samples for training. The methodology was validated using few-shot learning with CONCH-guided sample selection and evaluated on three publicly available datasets: (i) BACH (normal vs benign vs in situ vs invasive), (ii) MHIST (polyp classification), and (iii) SiCap (Gleason grading). Overall, this paper is well-written with a detailed literature review. An important finding from the results is that CONCH significantly outperforms QuiltNet, highlighting the significance of clean data for model training. Another key contribution is that the study demonstrates that fine-tuning a model up to a specific threshold (number of samples) can achieve performance comparable to few-shot supervised models using proposed annotation-free approach, though it is not discussed if performance is clinically meaningful.

# Weaknesses:

1.	While the study presents valuable contributions, several areas could benefit from further clarification and analysis. It would be helpful if the authors could provide more detailed information about the specific datasets from existing histopathology databases used for DAPT and TAPT training to enhance reproducibility.
2.	The rationale for sampling high-alignment score patches based on the CONCH model could be better explained, as this step can potentially exclude relevant high-quality pairs. Additional details about the selection criteria would strengthen this section.
3.	The opposing performance trends between QuiltNet and CONCH suggest there may be inherent sampling considerations when using CONCH for sample selection, and the high standard deviations across different scales make visual comparison challenging (figure 3).
4.	While the curves in Figure 3 show apparent differences, statistical significance testing would strengthen the conclusions, particularly given the high standard deviations observed in Figure 2.
# Detailed comments:
1.	Please provide more detailed information about the specific datasets from existing histopathology databases that were used for training with DAPT and TAPT to enhance reproducibility.
2.	Please explain the rationale for sampling high-alignment score patches based on the CONCH model. Also, please define the threshold used for this.
3.	Potential challenges with Equation 2's contrastive loss could have been explained in limitations when multiple images share identical textual descriptions, especially in cases with similar morphological regions, while models will try to push them away from each other.
4.	The opposing performance trends between QuiltNet and CONCH suggest there may be inherent sampling considerations when using CONCH for selection. Additionally, the high standard deviations across different scales make visual comparison challenging. Please consider normalization or alternative presentation methods.
5.	Please provide additional supporting references or justification for the explanation that "QuiltNet has already been pre-trained on Quilt1M, making domain information from additional caption pairs redundant, whereas CONCH, lacking this specific domain exposure, benefits more directly from DAPT."
6.	While the curves in Figure 3 show performance gains, please consider showing standard deviations to strengthen the conclusions,
7.	Please clarify the rationale for conducting Figure S1 experiments using QuiltNet for PathologyBERT and LoRA integration, considering its lower baseline performance compared to CONCH.
8.	Please provide a more comprehensive interpretation of Figures 2 and 3, particularly addressing the performance trajectory beyond the presented results in figure 3. Since TAPT performance shows a declining trend in Figure 2 and CoOp results are not presented after 100 samples, it’s unclear if we can obtain the same performance of CooP using TAPT by simply increasing the number of samples beyond 100 as 60% is not clinically meaningful.